# Exploring the Role of IL-17A in Oral Dysbiosis-Associated Periodontitis and Its Correlation with Systemic Inflammatory Disease

**DOI:** 10.3390/dj11080194

**Published:** 2023-08-12

**Authors:** Koichiro Irie, Tetsuji Azuma, Takaaki Tomofuji, Tatsuo Yamamoto

**Affiliations:** 1Department of Preventive Dentistry and Dental Public Health, Kanagawa Dental University, Yokosuka 238-8580, Japan; yamamoto.tatsuo@kdu.ac.jp; 2Department of Community Oral Health, School of Dentistry, Asahi University, Mizuho 501-0296, Japan; tetsuji@dent.asahi-u.ac.jp (T.A.); tomofu@dent.asahi-u.ac.jp (T.T.)

**Keywords:** oral dysbiosis, periodontitis, systemic inflammatory disease, interleukin-17A, immune response

## Abstract

Oral microbiota play a pivotal role in maintaining homeostasis, safeguarding the oral cavity, and preventing the onset of disease. Oral dysbiosis has the potential to trigger pro-inflammatory effects and immune dysregulation, which can have a negative impact on systemic health. It is regarded as a key etiological factor for periodontitis. The emergence and persistence of oral dysbiosis have been demonstrated to mediate inflammatory pathology locally and at distant sites. The heightened inflammation observed in oral dysbiosis is dependent upon the secretion of interleukin-17A (IL-17A) by various innate and adaptive immune cells. IL-17A has been found to play a significant role in host defense mechanisms by inducing antibacterial peptides, recruiting neutrophils, and promoting local inflammation via cytokines and chemokines. This review seeks to present the current knowledge on oral dysbiosis and its prevention, as well as the underlying role of IL-17A in periodontitis induced by oral dysbiosis and its impact on systemic inflammatory disease.

## 1. Introduction

The oral cavity serves as a primary gateway to the human body, where oral microorganisms play a pivotal role in maintaining a delicate balance between health and disease, both locally and systemically [1,2]. These microorganisms comprise the second largest microbiota community in the human body, consisting of various taxa, including bacteria, fungi, archaea, viruses, and protozoa [3]. The oral microbiome is composed of microbial groups that inhabit different ecological niches, such as saliva, tongue, dental surfaces, gingiva, buccal mucosa, palates, and tonsils [4]. These microorganisms exhibit spatial organization and form diverse biofilms that vary based on the aggregation of bacterial taxa [5]. In a healthy individual, the oral microbiome may harbor approximately 100–200 species of >700 oral microorganisms [6]. However, inter-individual variations are influenced by several factors, such as environmental conditions, genetic makeup, age, lifestyle, dietary habits, and oral hygiene [7].

The diversity of oral bacteria plays a pivotal role in upholding the stability and efficacy of the ecosystem [8]. Microbial diversity and stability are common key indicators of oral health given their inverse association with chronic diseases and metabolic dysfunction [9]. Oral bacteria primarily exist in the form of biofilms [10], and play an indispensable role in maintaining homeostasis, safeguarding the oral cavity, and impeding disease progression [11]. Pathogenic bacteria contribute to inflammatory diseases by remodeling normally benign bacteria, which results in an imbalance between normal and pathogenic bacteria (dysbiosis) [1,12]. The onset of inflammation is not solely attributable to one type of pathogenic bacteria; rather, it arises from alterations in the collective bacterial population [1,13]. *Porphyromonas gingivalis* (*P. gingivalis*), a Gram-negative bacterial pathogen, is one of the foremost causes of periodontitis. *P. gingivalis*, despite its low abundance, plays a significant role as a pathogen that perturbs the homeostasis of the commensal microbiota, resulting in a state of dysbiosis. Oral dysbiosis might trigger pro-inflammatory effects and immune dysregulation related to various oral diseases, including periodontitis, dental caries, endodontic infections, and oral cancer [14,15]. The dysbiosis of the human oral microbiota constitutes a pivotal etiological determinant of periodontitis [16,17]. This condition modifies the competitive balance among bacterial taxa by influencing the availability of nutrients and by inducing alterations in gene expression patterns within the microbial community. Such alterations may potentially bolster the pathogenicity of certain bacterial strains and, in turn, trigger inflammatory pathways [1,18]. Therefore, oral dysbiosis induces inflammation and disturbs the regulation of inflammatory responses, which leads to periodontitis. However, the mechanisms underlying the inflammatory response mediated by high cytokine expression induced by oral dysbiosis remain unclear.

Periodontitis is a chronic inflammatory disease initiated by oral bacteria that causes progressive destruction of tooth-supporting tissues, including the gingiva, periodontal ligament, and alveolar bone [19]. The host’s reaction to bacterial offense results in gingival swelling, bleeding upon gentle probing, elevated periodontal pocket depth, and alveolar bone loss. Periodontitis is related to important systemic diseases, including diabetes mellitus [20,21], cardiovascular disease [9,22] respiratory disease [23,24], rheumatoid arthritis [25,26], adverse pregnancy outcomes [22,27], cancer [28,29], and osteoporosis [30,31]. Therefore, there is a need to elucidate the mechanisms underlying symptoms of periodontitis induced by oral dysbiosis, their functional significance to the disease, and strategies for maintaining a healthy oral microbiota.

The interleukin-17 (IL-17) cascade has been linked to the development of environmental conditions that promote microbial dysbiosis. In addition, recent research has examined how IL-17A can lead a shift in the microenvironment towards highly pathologic bacterial settings, which can enhance periodontal inflammation. Furthermore, IL-17A is emerging as a major player in immune responses and inflammatory diseases. IL-17A target molecules include chemokines, cytokines (tumor necrosis factor-alpha (TNF-α), IL-1β, IL-6, and Granulocyte Macrophage colony-stimulating Factor (GM-CSF) and Receptor Activator of NFκB Ligand (RANKL). Additionally, it has been reported that the combined action of IL-17A with other cytokines, such as TNF-α, IL-1β, and Interferon (IFN)-γ, results in a synergistic amplification of pro-inflammatory responses. Therefore, IL-17A has an important role in the progression of periodontitis [1,32]. IL-17A is secreted by a variety of innate and adaptive immune cells and activates a series of inflammatory cascade responses that mediate the onset and development of periodontitis and systemic inflammatory diseases [33,34]. Moreover, IL-17A promotes the production of antimicrobial factors crucial for the containment of pathogens and commensal bacteria at barrier sites. Other immune cells, such as γδ T cells, natural killer (NK) cells, and type 3 innate lymphoid cells (ILC3s), that also produce IL-17A are strategically located at barrier sites and are the first line of defense for controlling neutrophil recruitment and production [35]. In periodontal pockets of patients with periodontitis, the number of γδ T cells and neutrophils, as well as the expression of IL-17A, are higher than in healthy individuals [36]. Thus, it is important to understand the mechanism of oral dysbiosis, its functional relationship to disease, and strategies for reversing oral dysbiosis and restoring health. In this review, we summarize the recent knowledge regarding oral dysbiosis and its prevention; further, we focus on the role of IL-17A as a pathophysiological mediator in the mechanisms linking periodontitis and systemic inflammatory diseases.

## 2. Role of Oral Bacteria in Maintaining Periodontal Tissue Homeostasis

Oral bacteria exert significant benefits to the host by inhibiting pathogen colony formation and affecting cell structure and normal development of the immune system [37]. Studies on germ-free (GF) mice have shown that bacterial colony formation is not essential for life. However, it is essential for health maintenance and contributes to the regulation of host responses related to health and disease [38,39]. In the oral cavity, the immune system maintains homeostasis by balancing the interaction of oral bacteria with the oral epithelium, as well as the constant tissue damage [4]. We previously showed that oral bacteria contribute to structural and functional changes in the oral area [40,41]. Furthermore, oral bacterial colonization limits the available binding sites and nutrients in the oral cavity for potential pathogens, which is referred to as “colonization resistance” [42]. In the context of the gut microbiome, it was observed that commensal bacteria contribute to healthy homeostasis through their immunostimulatory effect. This comprises the recruitment of immune cells in the mucosal lining, the creation and development of structured lymphoid tissues [43], and the induction of defensive capabilities in epithelial cells, encompassing the secretion of mucus and the synthesis of antimicrobial peptides [44]. Therefore, oral bacteria may exert functional effects as an additional host “tissue”, which contributes to homeostasis maintenance during health; further, impairment of this function may cause disease [45]. The innate barrier function of the epithelium is crucially involved in maintaining oral health. The epithelium of soft tissues, especially the junctional epithelium, is proximal to the highly vascularized gingiva, which provides a constant supply of innate immune cells to the gingival crevice for patrolling and controlling the bacterial load [37,46]. In healthy gingival tissues, the junctional epithelium remains in a low-differentiation state, with a turnover rate of approximately 4–6 days [47]. Compared with GF mice, conventional mice have a significantly increased junctional epithelium area [37]. Taken together, oral bacteria significantly influence periodontal tissue development and the regulation of immune responses to pathogens.

### Effects of Oral Bacteria on Innate Defense System

Studies comparing conventional and GF mice have revealed that bacterial colonization affects neutrophil regulation [37,41]. Additionally, following the administration of lipopolysaccharides (LPSs) derived from *P. gingivalis* to the gingival tissue, GF mice showed a lower number of CD4^+^ T cells in the periodontal tissues, as well as lower expression of TNF-α and fork head box protein p3 than specific pathogen-free mice [48]. The presence of oral bacteria in periodontal tissues induces an appropriate immune response through a highly synchronized process of neutrophil recruitment to gingival tissues, which is involved in monitoring bacterial colony formation and growth. Furthermore, the response to bacterial endotoxins by certain periodontal pathogens initiates an immune response that includes both innate immunity (macrophages, dendritic cells, NK cells, monocytes, neutrophils, etc.) and adaptive immunity (concerted action of B and T lymphocytes). Both of them are leading to production and release of inflammatory molecules, such as IL-17A, IFN-γ, IL-1β, and IL-6 [49]. Innate immunity and adaptive immunity are intricately intertwined. Innate immunity assumes the role of the initiator for adaptive immunity, facilitating the provision of activation signals essential for an effective adaptive immune response. For instance, Th cells secrete IFN-γ, a potent stimulator that robustly activates macrophages and NK cells, augmenting their phagocytic and cytotoxic capabilities [35]. Taken together, oral bacteria contribute to the highly organized expression of the host’s innate defense system, and especially neutrophils are expressed from the vasculature through the gingival tissue and ultimately into the gingival sulcus, forming a protective barrier between the gingival epithelium and oral bacteria.

## 3. Overview of Human Oral Dysbiosis

Dysbiosis refers to changes in the characteristic microbial communities of a specific microenvironment [50]. It can be characterized by the following three scenarios, which may simultaneously occur: (1) overall loss of microbiological diversity, (2) loss of beneficial microbes, and (3) expansion of pathogenic microbes [51]. Periodontal microbiome dysbiosis is clearly associated with periodontitis. Pathogenic bacteria are a minor component of the subgingival microflora in health, but increase with the development of periodontal pockets. Interestingly, dysbiotic communities were less diverse, but more similar among each other than healthy periodontal pocket communities [52]. Elucidating the mechanism underlying oral dysbiosis may inform the prevention of the transition from symbiosis to dysbiosis in oral bacteria. Oral dysbiosis mainly results from poor oral health behaviors, including poor oral care [53]. In periodontal tissue, biofilm accumulation causes an inflammatory response that increases gingival crevicular fluid (GCF) flow, which could be accompanied by bleeding. Increased gingival sulcus fluid flow not only induces a host defense response, but also serves as a substrate for many of the biased anaerobic and proteolytic bacteria present in gingival sulcus biofilms [54]. The other factors that cause oral dysbiosis include poor systemic health, unhealthy dietary habits, smoking, immune defects, genetic differences, and salivary gland dysfunction (Figure 1) [11,55,56,57]. Furthermore, a dysbiotic shift and microbiota imbalance result in the formation of a biofilm microbial community [14]. Therefore, human oral diseases may be caused by oral bacterial variations, rather than exogenous infections [58].

### 3.1. Human Oral Dysbiosis in Periodontal Tissue

Periodontitis is a complex disease related to the microbial community that is initiated by a microbiota change from a healthy symbiosis to a pathogenic dysbiosis [2]. The transition from periodontal tissue health to disease is a dramatic transition from a symbiotic microbial community (composed primarily of commensal bacterial genera, such as *Actinomyces* and *Streptococcus*) to a biodiverse microbial community structure composed primarily of anaerobic genera, such as *Bacteroides*, *Porphyromonas*, *Treponema*, and *Prevotella* [1]. This comprises the mechanism underlying the formation of oral dysbiosis; additionally, the microbial composition develops in stages within the gingival sulcus. Early bacterial colonizers of gingival tissue are only mildly pathogenic and are Gram-positive [59]. Beginning with salivary protein deposition on the tooth, bacterial adhesives and nutrient requirements are involved in oral biofilm development [60]. Upon the formation of biofilms, they grow and form increasingly complex structures. Gingivitis is caused by these biofilms located in or close to the gingival sulcus. The genera *Streptococcus*, *Fusobacterium*, *Actinomyces*, *Veillonella*, *Treponema*, *Bacteroides*, and *Capnocytophaga* are strongly implicated in gingivitis [61]. The development of periodontal dysbiosis occurs over a broadened timeframe, which slowly turns the symbiotic association of the host and the microbe to a pathogenic form. Among the microbial communities, the initial complex that has been associated with disease is the orange complex, comprised of anaerobic Gram-negative species, such as *Prevotella intermedia* (*P. intermedia*), *Prevotella nigrescens* (*P. nigrescens*), *Prevotella micros* (*P. micros*), and *Fusobacterium nucleatum* (*F. nucleatum*). As disease progresses, this complex transitions into the red complex, encompassing *Treponema denticola* (*T. denticola*), *Tannerella forsythia* (*T. forsythia*), and *P. gingivalis* [14]. Red complex exhibited a very strong relationship with pocket depth and bleeding on probing.

The two complexes are composed of species considered to be the main causative agents of periodontitis. The orange complex pathogens are known to have the ability to attach to several oral bacteria. These bacteria can bind to other bacteria and are generally considered to be organisms that bridge the resident colony that is periodontal pathogens [62]. The presence of this orange complex bacteria is very important, without which the aggressiveness of the red complex bacteria cannot survive in the oral cavity. They are highly pathogenic bacteria that are primarily Gram-negative and contain endotoxins [16]. The red complex bacteria can cause the proliferation of indigenous microflora, which increases the microbial load, inflammatory pathologies, and tissue destruction. Particularly, during the phase of limited colonization, *P. gingivalis* intricately perturbs the innate immune response and instigates alterations in the abundance and configuration of the oral commensal bacteria. Consequently, the resultant dysbiotic microbial assemblage impairs the equilibrium between the host and microbes, triggering inflammatory bone destruction. These observations imply that *P. gingivalis* can be categorized as a pivotal pathogenic organism [62]. Therefore, *P. gingivalis* play a critical role as a keystone pathogen in remodeling the commensal microbiota to a state of dysbiosis, even at low levels of abundance [16]. In addition, *P. gingivalis* possesses the ability to exert influence on adaptive immune responses through its selective facilitation of the differentiation and mobilization of Th17 cells, which represent a specific subset of T cells known for their inherent capacity for homeostasis, but are also strongly implicated in the progressive degradation of periodontal tissues [63].

Recent studies have disclosed emerging new pathogens, such as *Filifactor alocis* (*F. alocis*) and *Peptoanaerobacter stomatis* (*P. stomatis*), Gram-positive anaerobes that may play a significant role in periodontitis. There is increasing evidence to suggest that these bacteria play an important role in community dynamics, which thereby could be a major player causing dysbiosis [64]. When juxtaposed with other Gram-positive bacteria residing in the oral cavity, the profound alterations provoked in the host proteome as a consequence of *F. alocis* synergism have the potential to elicit numerous systemic host reactions [64]. *F. alocis* exhibits the capacity to intrude upon gingival epithelial cells, generate trypsin-like proteases, withstand oxidative stress, and exert control over neutrophil immune responses [65]. *P. stomatis* facilitates not only the migration of neutrophils, but also monocytes, thereby significantly exacerbating inflammation, coupled with the pronounced induction of granule content exocytosis [66]. The dysbiotic microbiome, in such a state, serves as a pathogenic determinant, rather than a mere consequence of the altered environment within this inflammatory condition.

### 3.2. Interaction between Oral Dysbiosis and the Junctional Epithelium

Oral dysbiosis modulates the junctional epithelial cells during infection and contributes to host cell signaling, metabolic host responses, and cell–cell interactions in oral bacteria [67]. Increased levels of LPS, an outer membrane component of Gram-negative bacteria, such as *P. gingivalis*, *F. nucleatum*, and *Aggregatibacter actinomycetemcomitans*, in the junctional epithelium stimulate the production of inflammatory mediators and cytokines, thereby promoting inflammation [68]. LPS triggers the secretion of several pro-inflammatory cytokines, including TNF-α, IL-1β, and IL-6, by junctional epithelial cells [69]. Therefore, oral dysbiosis induces neutrophil recruitment; stimulates the immune responses of cells, such as macrophages and dendritic cells; and elicits the release of inflammatory mediators to control Th cells [70]. This leads to heightened secretion of proinflammatory agents, such as IL-1β, TNF-α, and prostaglandin E2, thereby exacerbating the host’s response to microbes and resulting in systemic infection [71]. Consequently, if the infection persists, there is a continual discharge of pro-inflammatory agents that stimulate adaptive immunity by activating B and T cells [36].

## 4. Oral Dysbiosis in Periodontitis and Systemic Disease

Oral dysbiosis can directly trigger systemic inflammation, either through the escalation of inflammation caused by the release of toxins or the transportation of microbial products into the bloodstream [72]. Typically, oral dysbiosis instigates an immune response that involves both innate (neutrophils, macrophages, and dendritic cells) and adaptive immunity (T and B cells), which leads to the secretion of pro-inflammatory molecules, including interferon [IFN]-γ, IL-17A, TNF-α, IL-1β, IL-6, and related enzymes, particularly collagenases, like matrix metalloproteinases [1,62]. Therefore, oral dysbiosis in periodontitis has a causal relationship with the development and progression of systemic inflammatory diseases, such as diabetes mellitus, rheumatoid arthritis, and cardiovascular diseases, since numerous cytokines secreted during low-grade inflammation circulate from blood vessels throughout the body [1,9,12,22,26,30,53,72,73]. Numerous cytokines have a close association with specific T lymphocytes (naive CD4^+^ T cells) that are incited by inflammatory cytokines to differentiate into Th1, Th2, Th17, follicular helper T cells, and Treg cells [74]. Among these, Th1 (IL-12 and IFN-γ) and Treg (IL-2 and TGF-β, IL-10 family) cells manifest pleiotropic and anti-inflammatory effects in periodontitis. Additionally, Th17 (IL-17A and IL-23) and Th2 (IL-4, IL-5, IL-13) are related to pleiotropic effects [75]. The equilibrium of the Th17/Treg ratio plays a crucial role in preserving oral mucosal homeostasis [33]. Imbalanced homeostasis between Th17 and Treg has been attributed to periodontitis [76], thereby necessitating a harmonious balance between Th17 effector cells and Treg to maintain functional immunity and host health [77]. Tregs are known for their suppressive role in the immune system, accomplished through the release of cytokines or by direct cellular contact. These cells are vital for maintaining autoimmune tolerance. Interestingly, there have been reports of Tregs undergoing transdifferentiation into IFN-γ-secreting cells akin to Th1, and from IL-17A-producing Th17 cells into IFN-γ-secreting cells, which has been linked to the development of autoimmune diseases [78]. Notably, exFoxp3Th17 cells—Th17 cells derived from Foxp3^+^ T cells—have been found to have strong pro-inflammatory and pro-osteoclastogenic properties that promote the pathogenesis of autoimmune arthritis [79]. Moreover, these cells exhibit heightened osteoclastogenic activity compared to conventional Th17 cells, underscoring their crucial role in osteoclastogenesis. Th17 cells, which have been associated with various immune-mediated inflammatory conditions, such as psoriasis, rheumatoid arthritis, multiple sclerosis, inflammatory bowel diseases, and asthma, are potent mediators of tissue inflammation [34]. It is crucial to take into account the secretion of IL-17A when comprehending systemic inflammation, as diverse cellular entities, such as osteoblasts, endothelial cells, epithelial cells, chondrocytes, fibroblasts, keratinocytes, and macrophages, have the capability to express the IL-17A receptor [75]. IL-17A contributes to neutrophils leaving the bone marrow, entering blood circulation, and reaching the infected site of periodontitis [80]. Additionally, IL-17A plays a major protective role against bone loss resulting from *P. gingivalis*-induced periodontal disease; however, numerous studies have demonstrated that IL-17A is associated with bone erosion in rheumatoid arthritis [81]. Consequently, IL-17A collaborates with other pro-inflammatory cytokines, and the general maladjustment of these signaling molecules may have a hand in altering the host’s microbiome in the oral cavity, which could play a pivotal role in the development of systemic diseases. Therefore, this section focuses on the pathogenesis of systemic chronic inflammatory diseases related to IL-17A and induced by oral dysbiosis in periodontitis.

### 4.1. Role of IL-17A Production Induced by Oral Dysbiosis in Systemic Inflammatory Disease

IL-17A assumes a vital function in both physiological and pathological states. As an instance, IL-17A is promptly synthesized in reaction to microbial incursions by bacteria, fungi, and viruses. In chronic inflammatory disease, IL-17A is known to exert pro-inflammatory effects and induce pathological inflammatory responses that promote disease development, onset, and chronicity [75]. IL-17A activates numerous tissue- or cell-type-specific genes. IL-17A potently induces barrier host defense against microbes sensitive to neutrophils and antimicrobial protein activity, including periodontal pathogens and other disease pathogens. Contrastingly, IL-17 can drive tissue repair, as evidenced by some of the genes it regulates [82]. IL-17A is secreted by various innate and adaptive immune cells, such as NK cells, ILC3s, neutral regulatory T cells, γδT cells, and Tc17 cells [34]. These cells are crucially involved in the initiation and progression of psoriasis, rheumatoid arthritis, diabetes mellitus, and other disorders [1,83] (Figure 2). There is increasing interest in the association between periodontitis and chronic systemic inflammatory diseases, with numerous therapeutic interventions, including cytokine-based treatment strategies, improving periodontitis and systemic health [84]. Recently, genetic or pharmacological inhibition of Th17 cells or IL-17A can suppress periodontal bone loss, suggesting that these cells and related pathways could be applied in therapeutic interventions [85,86].

### 4.2. Periodontitis, IL-17A, and Their Association with Diabetes Mellitus

Diabetes mellitus (DM) is a metabolic disorder characterized by chronic hyperglycemia arising from inadequate insulin secretion, insulin resistance, or both [87]. Type 2 DM (T2DM), also known as non-insulin-dependent DM, is the most prevalent subtype of DM, and is primarily caused by a combination of insulin resistance and impaired insulin secretion [87]. Periodontitis and DM exhibit bidirectional interactions and reciprocal associations [88]. Hyperglycemia is closely linked to the development and severity of periodontitis [89]. The immune inflammatory response induced by periodontitis could lead to insulin resistance; further, periodontitis can increase DM severity [90]. Notably, DM reduces the diversity of gut microbiota, which is consistent with reports that DM decreases bacterial diversity in other tissues [91]. In the oral cavity, irrespective of periodontal health status, diabetic patients manifest a greater number of oral microbial traits, such as microbial composition, biological diversity, and the relative abundance of specific bacteria, when compared to healthy controls [92]. The subgingival microbiome of diabetic patients with healthy periodontium is characterized by reduced species richness compared to non-diabetic healthy individuals, with relatively high abundance of opportunistic pathogenic orange complex species and the red complex species, and fewer species that are compatible with periodontal health [93]. A shift towards a more pathogenic bacterial profile in the oral microbiota may contribute to an increased risk of periodontitis in patients with diabetes mellitus. Notably, diabetes mellitus is known to reduce the diversity of oral microbiota, which is consistent with reports indicating that it decreases bacterial diversity in other tissues [91]. Oral administration of *P. gingivalis* in mice causes simultaneous development of systemic inflammation and insulin resistance [94]. Furthermore, patients with diabetes mellitus and chronic periodontitis have increased salivary levels of IL-17A compared with healthy controls, suggesting that elevated IL-17A levels are a risk factor for chronic periodontitis in these patients [95]. Taken together, oral dysbiosis could influence the bidirectional relationship between periodontitis and DM; moreover, IL-17A expression resulting in oral floral changes induced by increased systemic inflammation may increase insulin resistance and promote hyperglycemia.

### 4.3. Periodontitis, IL-17A, and Their Association with Atherosclerosis

Atherosclerosis is a persistent inflammatory ailment of the vascular system that is distinguished by the production of an atherosclerotic plaque within the walls of medium- or large-sized elastic muscular arteries; furthermore, it is the primary etiology of cardiovascular disease [96]. Periodontitis represents a crucial risk factor for the onset of atherosclerosis [97]. Periodontal pathogens, including *P. gingivalis*, *T. forsythia*, and *P. intermedia*, can directly enter atherosclerotic lesions and are involved in the plaque formation process [98]. In addition, *P. gingivalis* can induce oxidation of low-density lipoprotein, which causes atherosclerosis [99]. Furthermore, *P. gingivalis* can cause arterial endothelial dysfunction, foam cell formation, vascular smooth muscle cell proliferation and calcification, and Th cell–Treg imbalance [100]. Moreover, *P. gingivalis* and its LPS have the ability to stimulate monocytes; facilitate Th17/IL-17A responses; elevate levels of TNF-α, IL-1β, IL-6, and IL-17A; and induce atherosclerotic plaque formation via TLR2/TLR4 signaling and inflammatory responses [33]. IL-17A expression is upregulated in atherosclerosis and pathophysiologically contributes to atherosclerosis by increasing plaque size and aggravating inflammation [101]. Taken together, oral dysbiosis frequently causes bacteremia, which damages the endothelial tissue; furthermore, bacteria-induced LPS production and release may chronically activate circulating immune cells to promote atherosclerosis.

### 4.4. Periodontitis, IL-17A, and Their Association with Rheumatoid Arthritis

Rheumatoid arthritis (RA) is a persistent, immune-mediated, inflammatory disease that affects the synovial membrane and causes systemic inflammation [102]. Moreover, periodontitis represents a risk factor for the development of RA [103,104], as RA and periodontitis share common pathogenetic and immunological features, such as heightened inflammation, immune cell infiltration, augmented secretion of comparable cytokines and pro-inflammatory agents, reduced release of anti-inflammatory mediators, and activation of the NF-κB/RANKL signaling pathway [105]. RANKL is secreted by Th17 cells stimulated by IL-23 or by fibroblasts stimulated by IL-17A [106]. Synovitis is a multifaceted signaling system that encompasses immune cells and cytokines. This network becomes vascularized and infiltrated by fibroblasts, macrophages, T cells, B cells, plasma cells, mast cells, dendritic cells, and neutrophils [107]. Additionally, LPS-stimulated dendritic cells and macrophages produce IL-23, which binds to the specific receptor IL-23R on Th17 cells, promoting and maintaining clonal expansion, and triggering the secretion of IL-17A and RANKL [108]. Furthermore, serum IL-17A levels in RA patients with periodontitis are significantly higher than those in periodontitis patients without systemic disease [109]. Interestingly, *P. gingivalis* is the only known bacterium that expresses peptidylarginine deiminase, which is involved in RA pathogenesis [110]. Taken together, periodontitis is an imbalanced subgingival community and host immune response, resulting in a dysbiotic condition. It is also a chronic autoimmune inflammatory disease, causing synovial inflammation and hyperplasia, which results in irreparable damage to the joints’ cartilage and bone. Therefore, alterations in IL-17A levels induced by periodontitis may systemically affect synovial tissue destruction and influence RA pathogenesis.

### 4.5. Periodontitis, IL-17A, and Their Association with Adverse Pregnancy Outcomes

There is a high prevalence of periodontitis among pregnant women. Hormonal and immune changes during pregnancy cause significant changes in oral bacteria. Maternal periodontitis increases the risk of adverse pregnancy events, including low birth weight, premature delivery, miscarriage, and stillbirth [111,112]. There are elevated levels of inflammatory mediators in the gingival sulcus fluid of women with adverse pregnancy outcomes, which indicates that inflammatory cytokines may promote labor [113,114]. Additionally, compared with non-pregnant, pregnant women have an over-representation of the genera *Neisseria*, *Porphyromonas*, and *Treponema*, as well as an under-representation of *Streptococcus* and *Veillonella* [3]. Constitutional changes during pregnancy may increase the risk of infection by harmful oral bacteria, leading to oral dysbiosis, which may trigger disease. Hence, comparable to the pathogenesis of atherosclerosis and RA, inflammatory mediators, including TNF-α, IL-1β, IL-6, and IL-17A, generated in the context of periodontitis can enter the systemic circulation and trigger an acute-phase response that could negatively impact the placenta and fetus [115].

## 5. Prevention of Human Oral Dysbiosis

To prevent oral dysbiosis, it is important to recognize the pathogenic function of bacteria accumulating in the periodontal pocket. In periodontitis, there is an augmented diversity of oral microbiota, and the subgingival plaque is noted for its greatest richness and diversity of species within the oral cavity [116]. The primary approach to preventing oral dysbiosis and interfering with the local ecological niche of bacteria in both the supra- and subgingival regions is mechanical debridement of biofilms from the tooth surface [117]. There are sufficient data to support that individual plaque control techniques, such as conventional tooth brushing and the use of chemical plaque removers, can significantly improve gingival inflammation and lower plaque scores if cleaning is thorough enough and performed at appropriate time intervals [118]. However, their complete elimination is not the solution. Rather, techniques to regulate the oral environment and re-establish a better oral environment are desired. In recent years, systemic antibiotics [119], probiotics [120], and photodynamic therapy [121] might be useful to decrease the load of pathogenic bacteria, even though strong evidence-based conclusions could not been drawn yet. We believe that a central mechanism for altering the oral microbiota is an increase in IL-17A. In addition, we recognize that IL-17A acts in concert with other inflammatory cytokines and that overall dysregulation of these mediators may contribute to host changes leading to oral dysbiosis. Recently, it has been reported that continuous administration of IL-17A-neutralizing antibodies to the periodontal region can inhibit inflammatory bone loss in mice with experimental periodontitis in an animal study [122]. Further careful studies with IL-17A inhibitors are needed for application in clinical research. Thus, there is no established preventive strategy for oral dysbiosis. Evaluation of oral flora diversity and composition may inform the establishment of novel prevention methods and the evaluation of efficacy.

## 6. Conclusions

Oral dysbiosis is a key etiological factor for periodontitis. Therefore, elucidating the physiological and metabolic characteristics of oral dysbiosis is very important for finding novel treatments for periodontitis that regulate the activities of bacterial communities and bring them to a healthy state. Oral dysbiosis induces IL-17A expression in the periodontal tissue. Additionally, IL-17A synergistically interacts with other inflammatory cytokines, and the overall dysregulation of these mediators may participate in host alterations that lead to oral dysbiosis and play a pivotal role in the pathogenesis of DM, atherosclerosis, RA, and adverse pregnancy outcomes. Although IL-17A inhibition can effectively treat psoriasis [123] and RA [124] in the human study, IL-17A inhibitors have been rarely used to treat periodontitis. This could be attributed to the involvement of IL-17A in the progression of periodontal disease remaining unclear. IL-17A exerts both physiological and pathological effects, and its inhibitors have significant side effects. Consequently, the causality between some of these diseases and periodontitis remains enigmatic. Comprehending the role of oral dysbiosis in the pathogenesis, early prevention, and treatment of periodontitis may aid in the amelioration of systemic inflammatory ailments. In addition, understanding the diversity and composition of the oral microbiota may lead to the establishment of new preventive methods and evaluation of their efficacy.

## Figures and Tables

**Figure 1 dentistry-11-00194-f001:**
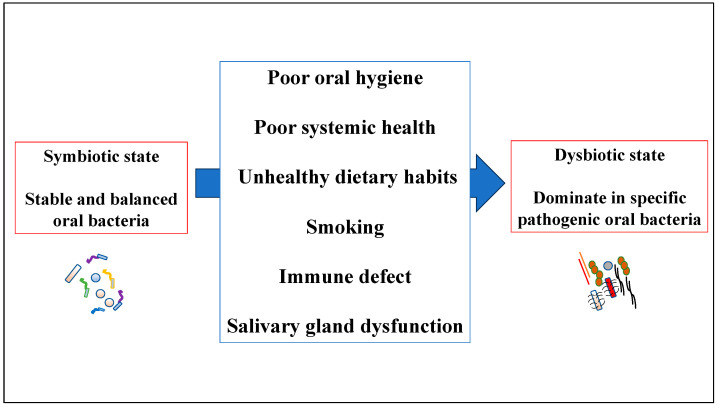
Various factors affecting the alteration of oral microbiome.

**Figure 2 dentistry-11-00194-f002:**
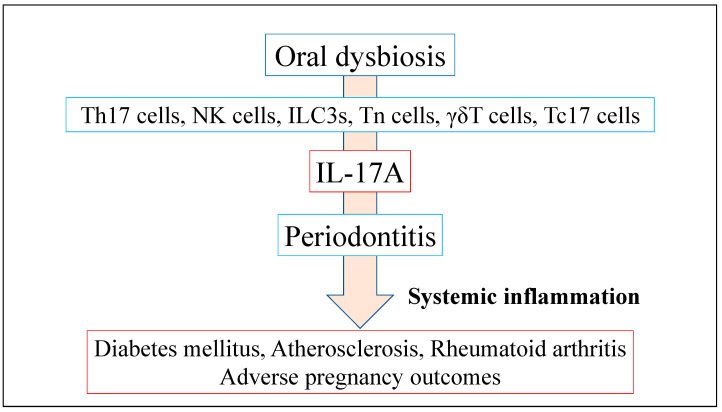
One possible mechanism of oral dysbiosis linking periodontitis to systemic inflammatory disease. Various types of innate and adaptive immune cells can secret IL-17A. Oral dysbiosis increases IL-17A expression, and contributes to host changes that also exacerbate oral dysbiosis, and plays a significant role in systemic inflammatory disease.

## Data Availability

Not applicable.

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
