# Peer review of "Exploring the Role of IL-17A in Oral Dysbiosis-Associated Periodontitis and Its Correlation with Systemic Inflammatory Disease"

_dentistry, 2023, doi:10.3390/dj11080194_

Round 1

Reviewer 1 Report

The present narrative review focuses on the role of oral dysbiosis in periodontal disease and the involvement of the IL-17 into the inflammatory process that, additionally, link periodontitis with some systemic diseases. However, this is not a novelty topic since it has been extensively investigated in the last years.

My comments related to the manuscript are the following:

1) "Systemic antibiotics [108], probiotics [109] and photodynamic therapy [110] can also be used to suppress pathogenic bacteria within oral cavity,  especially in periodontal pockets." (lines 332-334) This sentence should be modified since it has not been established yet that those therapies are 100% effectively in suppressing periodontal bacteria, but there are controversial results. Thus, expressions such as "might be useful to decrease the load of pathogenic bacteria...even though strong evidence based conclusions could not been drawn yer" should be used.

2) At the Conclusion section, be careful with these two sentences "Therefore, elucidating the physiological and metabolic characteristics of oral dysbiosis could inform novel treatments for periodontitis that regulate the activities of bacterial communities and bring them to a healthy state."(lines 343-344); "Good oral hygiene and avoidance of dysbiosis can effectively prevent periodontitis." (lines 356-357). Considering that there are also other important factors such as systemic factors and/or genetic factors, it should not be stated that with just a good oral hygiene and with the "avoidance" of the dysbiosis there will be a prevention of periodontitis, as well as it is not easy to find treatments that might regulate the whole activity of bacteria, to reach a fully healthy periodontal state. Therefore, my recommendation is to be more realistic with the conclusions, in order to give to the readers a right scientific message.

Author Response

1) "Systemic antibiotics [108], probiotics [109] and photodynamic therapy [110] can also be used to suppress pathogenic bacteria within oral cavity,  especially in periodontal pockets." (lines 332-334) This sentence should be modified since it has not been established yet that those therapies are 100% effectively in suppressing periodontal bacteria, but there are controversial results. Thus, expressions such as "might be useful to decrease the load of pathogenic bacteria...even though strong evidence based conclusions could not been drawn yer" should be used.

Our response: Thank you for your good suggestion. We agree your advice. We have revised the sentence based on your suggestions (lines 377- 390).

2) At the Conclusion section, be careful with these two sentences "Therefore, elucidating the physiological and metabolic characteristics of oral dysbiosis could inform novel treatments for periodontitis that regulate the activities of bacterial communities and bring them to a healthy state."(lines 343-344); "Good oral hygiene and avoidance of dysbiosis can effectively prevent periodontitis." (lines 356-357). Considering that there are also other important factors such as systemic factors and/or genetic factors, it should not be stated that with just a good oral hygiene and with the "avoidance" of the dysbiosis there will be a prevention of periodontitis, as well as it is not easy to find treatments that might regulate the whole activity of bacteria, to reach a fully healthy periodontal state. Therefore, my recommendation is to be more realistic with the conclusions, in order to give to the readers a right scientific message.

Our response: Thank you for your comments. To be more realistic with the conclusion, we deleted the sentence based on your suggestions and added some message at section 5 and 6.  

Reviewer 2 Report

Oral dysbiosis refers to an imbalance in the microbial community within the oral cavity, typically characterized by an overgrowth of harmful bacteria and a decrease in beneficial bacteria. It can have significant implications for oral health and overall well-being. Oral dysbiosis is closely linked to various oral health problems, such as periodontal disease. Furthermore, emerging research suggests oral dysbiosis may have systemic health implications beyond the oral cavity. There is growing evidence of associations between oral dysbiosis and systemic inflammatory diseases. The manuscript presents a comprehensive revision of the current knowledge on oral dysbiosis associated with periodontitis and discusses the role of IL-17A in systemic inflammatory disease. While its role in the gut has been extensively studied and described, its specific functions in the oral cavity are not as well characterized. However, research has shed some light on the potential involvement of IL-17A in oral health and diseases.

The authors have done excellent work on revising the available literature, but it might be beneficial to consider minor suggestions to strengthen the article.

Comments:

The authors discuss oral microbial dysbiosis around the Porphyromonas gingivalis as the keystone for periodontitis, but other important microbial participants should also be discussed. Periodontitis is a complex oral disease primarily caused by a dysbiotic oral microbiome, which involves the overgrowth of pathogenic bacteria and an imbalance in the microbial community. Plaque analysis from periodontitis-diseased sites has revealed the presence anaerobic bacterial complex consisting of the triad of P. gingivalisTannerella forsythia, and Treponema denticola (red complex). Furthermore, association with the disease was also linked to Prevotella spp., Fusobacterium spp. and Parvimonas micra (orange complex).

In addition, newly recognized periodontal pathogens should also be discussed. In recent years, new periopathogens such as Filifactor alocis and Peptoanaerobacter stomatis have been identified.

Lines 51-52, the references are suitable for periodontitis [2,11] and should add citations for the other referred oral diseases.

Line 82, “innate cells,” is too general; please provide examples of the major innate cell subsets linked to the Il-17 induction. Due to the complexity of different oral niches, maybe discuss innate immune cell location versus Il-17 induction. For example, the PMNs in the gingival pocket and the activation of the IL-17 pathway. 

Line 92, “However, it is essential for life”, seems duplicated; advice to remove it from the tex.

Line 99-102, Since the described role of commensal bacteria role healthy homeostasis refers to the gut, authors should say something like “In the context of the gut microbiome was observed that commensal bacteria….”

Subsection 2.1 (Effects of oral bacteria on innate immune defense system) needs to be developed more. Some other critical innate cells/mediators contribute to oral health. There is no link between the described innate immunity mediators to the Il-17 or the rest of the manuscript. Besides the incomplete innate immunity, it would make sense to have the effects of oral bacteria on adaptive immunity. 

Lines 147-148, add citations to each particular factor referenced.

In Section 3.1, as mentioned above for the introduction, authors should also discuss newly discovered periopathogens.

Line 183 add a citation.

Lines 192-193 add citations for the referred systemic inflammatory diseases or refer to that subject reviewed in citation [63].

In section 5. (Prevention of human oral dysbiosis) the authors should discuss the different mentioned treatments, approaches and respective limitations.  

Author Response

The authors discuss oral microbial dysbiosis around the Porphyromonas gingivalis as the keystone for periodontitis, but other important microbial participants should also be discussed. Periodontitis is a complex oral disease primarily caused by a dysbiotic oral microbiome, which involves the overgrowth of pathogenic bacteria and an imbalance in the microbial community. Plaque analysis from periodontitis-diseased sites has revealed the presence anaerobic bacterial complex consisting of the triad of P. gingivalisTannerella forsythia, and Treponema denticola (red complex). Furthermore, association with the disease was also linked to Prevotella spp., Fusobacterium spp. and Parvimonas micra (orange complex).

In addition, newly recognized periodontal pathogens should also be discussed. In recent years, new periopathogens such as Filifactor alocis and Peptoanaerobacter stomatis have been identified.

Lines 51-52, the references are suitable for periodontitis [2,11] and should add citations for the other referred oral diseases.

Our response: Thank you for your comments. We have changed the references (lines 51).

Line 82, “innate cells,” is too general; please provide examples of the major innate cell subsets linked to the Il-17 induction. Due to the complexity of different oral niches, maybe discuss innate immune cell location versus Il-17 induction. For example, the PMNs in the gingival pocket and the activation of the IL-17 pathway. 

Our response: Thank you for your comments. We have added the sentence based on your suggestions (lines 81-85).

Line 92, “However, it is essential for life”, seems duplicated; advice to remove it from the tex.

Our response: Thank you for your comments. We have removed it.

Line 99-102, Since the described role of commensal bacteria role healthy homeostasis refers to the gut, authors should say something like “In the context of the gut microbiome was observed that commensal bacteria….”

Our response: Thank you for your comments. We have added the sentence based on your suggestions (lines 99-100).

Subsection 2.1 (Effects of oral bacteria on innate immune defense system) needs to be developed more. Some other critical innate cells/mediators contribute to oral health. There is no link between the described innate immunity mediators to the Il-17 or the rest of the manuscript. Besides the incomplete innate immunity, it would make sense to have the effects of oral bacteria on adaptive immunity. 

Our response: Thank you for nice comments. We have added the sentence based on your suggestions (lines 122-131).

Lines 147-148, add citations to each particular factor referenced.

Our response: Thank you for your comments. We have added the references (lines 151).

In Section 3.1, as mentioned above for the introduction, authors should also discuss newly discovered periopathogens.

Our response: Thank you for nice suggestions. We have totally changed based on your suggestions (Section 3.1).

Line 183 add a citation.

Our response: Thank you for your comments. We have added the references (lines 228).

Lines 192-193 add citations for the referred systemic inflammatory diseases or refer to that subject reviewed in citation [63].

Our response: Thank you for your comments. We have changed the references (lines 239).

In section 5. (Prevention of human oral dysbiosis) the authors should discuss the different mentioned treatments, approaches and respective limitations. 

Our response: Thank you for your comments. We have changed the sentence based on your suggestions (lines 377-390).

Reviewer 3 Report

Comments:

Manuscript titled “Involvement of oral dysbiosis in periodontitis and its association with systemic inflammatory disease due to IL-17A expression.”

The review article from the authors, Irie K et al. have emphasized on the current knowledge on oral dysbiosis and its prevention, and the underlying role of IL-17A in periodontitis induced by oral dysbiosis and its impact on systemic inflammatory disease. Though the article is interesting, it can be comprehensive and has a scope of lot of improvement as mentioned below.

1. The manuscript title and schematics (Figure 1 and Figure 2) can be modified for better understanding.

2. The review is based on oral dysbiosis particularly, bacteria in periodontitis and association with systemic diseases but has only few mentions of the periodontal pathogens (bacterial names) in the whole text. The authors should have an inclusive overview of most periodontal pathogens mentioning their names and role in periodontitis and its impact on systemic diseases and crosstalk with immune response (IL-17A). In addition, ‘gingivitis’ causing bacteria which can lead to periodontitis should be discussed.

3. The sentences in the lines 91 and 92 are contradictory – ‘Studies on germ-free (GF) mice have shown that bacterial colony formation is not essential for life. However, it is essential for life…. ‘ – modify these for clarity.

4. The authors should explain in detail the bacterial communities involved in early colonization etc. instead of just using the terminologies throughout the article, such as early colonizers (Line 157).

5. Explain on the term ‘keystone pathogens’ - mention the names of keystone pathogens in periodontitis besides P. gingivalis in Line166 and in Figure 1.

6. The section ‘Prevention of human oral dysbiosis’ need a lot of improvisation expanding on the treatment options in detail.

7. Avoid repetitiveness of sentences and more recent references should be incorporated.

Decision: Major revision

Avoid repetition of sentences

Author Response

  1. The manuscript title and schematics (Figure 1 and Figure 2) can be modified for better understanding.

Our response: Thank you for your comments. We have revised the title and Figure based on your suggestions.

  1. The review is based on oral dysbiosis particularly, bacteria in periodontitis and association with systemic diseases but has only few mentions of the periodontal pathogens (bacterial names) in the whole text. The authors should have an inclusive overview of most periodontal pathogens mentioning their names and role in periodontitis and its impact on systemic diseases and crosstalk with immune response (IL-17A). In addition, ‘gingivitis’ causing bacteria which can lead to periodontitis should be discussed.

Our response: Thank you for nice suggestions. We have revised the sentence based on your suggestions (section 2.1 and 3.1).

  1. The sentences in the lines 91 and 92 are contradictory – ‘Studies on germ-free (GF) mice have shown that bacterial colony formation is not essential for life. However, it is essential for life…. ‘ – modify these for clarity.

 Our response: Thank you for your comments. We have removed it.

  1. The authors should explain in detail the bacterial communities involved in early colonization etc. instead of just using the terminologies throughout the article, such as early colonizers (Line 157).

Our response: Thank you for your comments. We have changed the sentence based on your suggestions (section 3.1).

  1. Explain on the term ‘keystone pathogens’ - mention the names of keystone pathogens in periodontitis besidesP. gingivalis in Line166 and in Figure 1.

Our response: Thank you for your comments. We have added the sentence based on your suggestions (lines 190-201).

  1. The section ‘Prevention of human oral dysbiosis’ need a lot of improvisation expanding on the treatment options in detail.

 Our response: Thank you for your comments. We have revised the sentence based on your suggestions (Line 377-390).

  1. Avoid repetitiveness of sentences and more recent references should be incorporated.

Our response: Thank you for your comments. Based on your comments, we have revised the sentences as much as possible and added new references.

Round 2

Reviewer 3 Report

The authors have addressed the most of the comments suggested in the revised manuscript. However, the Figure 1 title and graphics are still unclear and need to be revised and structured appropriately for better understanding. Lines 166 and 167, Firmicutes, Proteobacteria,... are not genera but bacterial phyla. The authors should proofread the manuscript thoroughly to avoid such major errors. The microbiota names should be italicized throughout the manuscript.

Decision: minor revision

Moderate editing of english language required to improvise the manuscript.

Author Response

Thank you for your comments. We revised Figure 1 title and checked the microbiota names.